# Angiopoietin receptor Tie2 is required for vein specification and maintenance via regulating COUP-TFII

Man Chu[1†], Taotao Li[1†], Bin Shen[1,2], Xudong Cao[1], Haoyu Zhong[1], Luqing Zhang[1,2], Fei Zhou[1,2], Wenjuan Ma[1], Haijuan Jiang[1], Pancheng Xie[3], Zhengzheng Liu[1], Ningzheng Dong[1,4], Ying Xu[3], Yun Zhao[1,4], Guoqiang Xu[5], Peirong Lu[6], Jincai Luo[7], Qingyu Wu[1,4], Kari Alitalo[8], Gou Young Koh[9], Ralf H Adams[10], Yulong He[1,3,4,11]*

[1]Cyrus Tang Hematology Center, Collaborative Innovation Center of Hematology, Soochow University, Suzhou, China; [2]MOE Key Laboratory for Model Animal and Disease Study, Model Animal Research Institute, Nanjing University, Nanjing, China; [3]Cam-Su Genomic Resources Center, Soochow University, Suzhou, China; [4]Jiangsu Institute of Hematology, The First Affiliated Hospital, Soochow University, Suzhou, China; [5]College of Pharmaceutical Sciences, Soochow University, Suzhou, China; [6]The First Affiliated Hospital, Soochow University, Suzhou, China; [7]Laboratory of Vascular Biology, Institute of Molecular Medicine, Peking University, Beijing, China; [8]Wihuri Research Institute and Translational Cancer Biology Center of Excellence, Biomedicum Helsinki University of Helsinki, Helsinki, Finland; [9]Center for Vascular Research, Institute of Basic Science and Korea Advanced Institute of Science and Technology (KAIST), Daejeon, Korea; [10]Max-Planck-Institute for Molecular Biomedicine, Department of Tissue Morphogenesis, Faculty of Medicine, University of Münster, Münster, Germany; [11]Jiangsu Key Laboratory of Preventive and Translational Medicine for Geriatric Diseases, Soochow University, Suzhou, China

*For correspondence: heyulong@suda.edu.cn

[†]These authors contributed equally to this work

**Abstract** Mechanisms underlying the vein development remain largely unknown. Tie2 signaling mediates endothelial cell (EC) survival and vascular maturation and its activating mutations are linked to venous malformations. Here we show that vein formation are disrupted in mouse skin and mesentery when Tie2 signals are diminished by targeted deletion of *Tek* either ubiquitously or specifically in embryonic ECs. Postnatal Tie2 attenuation resulted in the degeneration of newly formed veins followed by the formation of haemangioma-like vascular tufts in retina and venous tortuosity. Mechanistically, Tie2 insufficiency compromised venous EC identity, as indicated by a significant decrease of COUP-TFII protein level, a key regulator in venogenesis. Consistently, angiopoietin-1 stimulation increased COUP-TFII in cultured ECs, while Tie2 knockdown or blockade of Tie2 downstream PI3K/Akt pathway reduced COUP-TFII which could be reverted by the proteasome inhibition. Together, our results imply that Tie2 is essential for venous specification and maintenance via Akt mediated stabilization of COUP-TFII.

## Introduction

Mechanisms underlying arteriovenous specification have been under intensive investigation during the past years, and this has led to the identification of several signaling pathways involved in the coordination of this process. The VEGF-A/VEGFR-2 pathway mediates activation of RAF1 and ERK1/

2 kinases to induce the expression of genes required for arterial development (*Lanahan et al., 2013*; *Deng et al., 2013*), including Delta-like 4 (Dll4) that activates NOTCH signaling (*Lawson et al., 2001*; *Duarte et al., 2004*; *Wythe et al., 2013*). Wnt/β-catenin, SOX17 and FOXC1/2 were also reported to participate in arterial development via activation of the NOTCH pathway (*Seo et al., 2006*; *Corada et al., 2010*, *2013*). In contrast, knowledge on venogenesis is still limited. COUP-TFII, a transcription factor expressed in venous but not arterial endothelial cells (ECs), has been shown to regulate venous identity via the inhibition of NOTCH mediated signals (*You et al., 2005*). Vice versa, NOTCH activation has been shown to down-regulate COUP-TFII expression (*Swift et al., 2014*). Furthermore, Akt activation was shown to inhibit Raf1-ERK1/2 signaling in ECs to favor venous specification (*Ren et al., 2010*). To date, however, the specific factors upstream of the Akt pathway that define venous EC identity remain unclear.

Tie2 is a receptor tyrosine kinase that mediates angiopoietin signaling for EC survival, vascular remodeling and integrity (*Augustin et al., 2009*). Tie2 deficiency led to embryonic lethality resulting from the defective vascular remodeling and heart development (*Dumont et al., 1994*; *Sato et al., 1995*), and combined deletion of Tie2 ligands Ang1 and Ang2 in mice was also shown to disrupt Schlemm's canal formation leading to ocular hypertension and glaucoma (*Thomson et al., 2014*). Patients with venous malformations were shown to have Tie2 missense point mutations (*Vikkula et al., 1996*), leading to ligand-independent Tie2 activation (*Limaye et al., 2009*). However, the underlying mechanism of Tie2 function in the blood vessels has not been fully elucidated. In this study, we show that Tie2 deficiency or insufficiency induced by gene targeting leads to defective vein formation and maintenance during embryogenesis and postnatal development. Findings from this study suggest that Tie2 is essential for the specification of venous EC identity via the Akt mediated regulation of COUP-TFII protein stability.

## Results

### Disruption of vein development after Tie2 deletion during embryogenesis

To characterize Tie2 function in vascular development, a conditional knockout mouse model targeting the *Tek* gene (*Shen et al., 2014a*) was employed in this study. Ubiquitous deletion of *Tek* led to embryonic lethality by E10.5 (*Figure 1—figure supplement 1A–C*), as previously reported (*Sato et al., 1995*). As shown in *Figure 1A,B* and *Figure 1—figure supplement 1D,E*, no veins (arrows) were detected in the head or somite regions of the *Tek* null embryos at E9.5, unlike in the littermate control embryos. Interestingly, Tie2 expression in the E9.5 embryos was higher in veins than in arteries (arrowhead, *Figure 1B*). The lack of veins in the intersomitic regions of *Tek* deleted mice was also evident by Dll4 and PECAM-1 double staining (*Figure 1—figure supplement 1E*).

To investigate the role of Tie2 later during embryogenesis, we employed the *Ubc-Cre$^{ERT2}$* and *Cdh5-Cre$^{ERT2}$* deletor mouse lines (*Wang et al., 2010*), to generate the doubly transgenic mice (*Tek$^{Flox/-}$/Ubc-Cre$^{ERT2}$*, named Tek$^{-/iUCKO}$; or *Tek$^{Flox/-}$/Cdh5-Cre$^{ERT2}$*, named Tek$^{-/iECKO}$). *Tek$^{Flox/+}$/Ubc-Cre$^{ERT2}$* or *Tek$^{Flox/+}$/Cdh5-Cre$^{ERT2}$* littermate mice were used as controls. Tie2 deletion efficiency was examined by immunostaining with skin and also by real-time RT-PCR with lung of *Tek* mutant mice (Tek$^{-/iECKO}$: 0.33 ± 0.07, n = 3; Tek$^{+/iECKO}$: 1.0 ± 0.26, n = 3; Tek$^{-/iUCKO}$: 0.25 ± 0.02, n = 4; Tek$^{+/iUCKO}$: 1.0 ± 0.11, n = 4). Analysis of veins in skin at E15.5 showed that venogenesis was disrupted in both types of mutant embryos, in which *Tek* deletion was induced by intraperitoneal injections of tamoxifen into pregnant mice at E10.5–12.5 (*Figure 1C–E* and *Figure 1—figure supplement 2A*). In addition, defective vein formation was also observed in the skin of E17.5 Tek$^{-/iUCKO}$ embryos when *Tek* deletion was induced at E12.5–14.5 (*Figure 1F*). Lack of bleeding or edema with the later targeting suggests a stage-dependent function of Tie2 during the establishment of vascular integrity. Interestingly, veins were detected in mesentery, but unlike in littermate controls, they did not align properly with the arteries (*Figure 1—figure supplement 2B*). Furthermore, lymphatic vessels originate mainly from veins during embryonic development in mammals. It is therefore interesting to find out whether lymphatic development is altered in Tie2 knockout mice when the vein formation is defective. We found that lymphatic vessels were present in the skin of Tek$^{-/iECKO}$ mutant mice, but became dilated (*Figure 1—figure supplement 3*). This may be

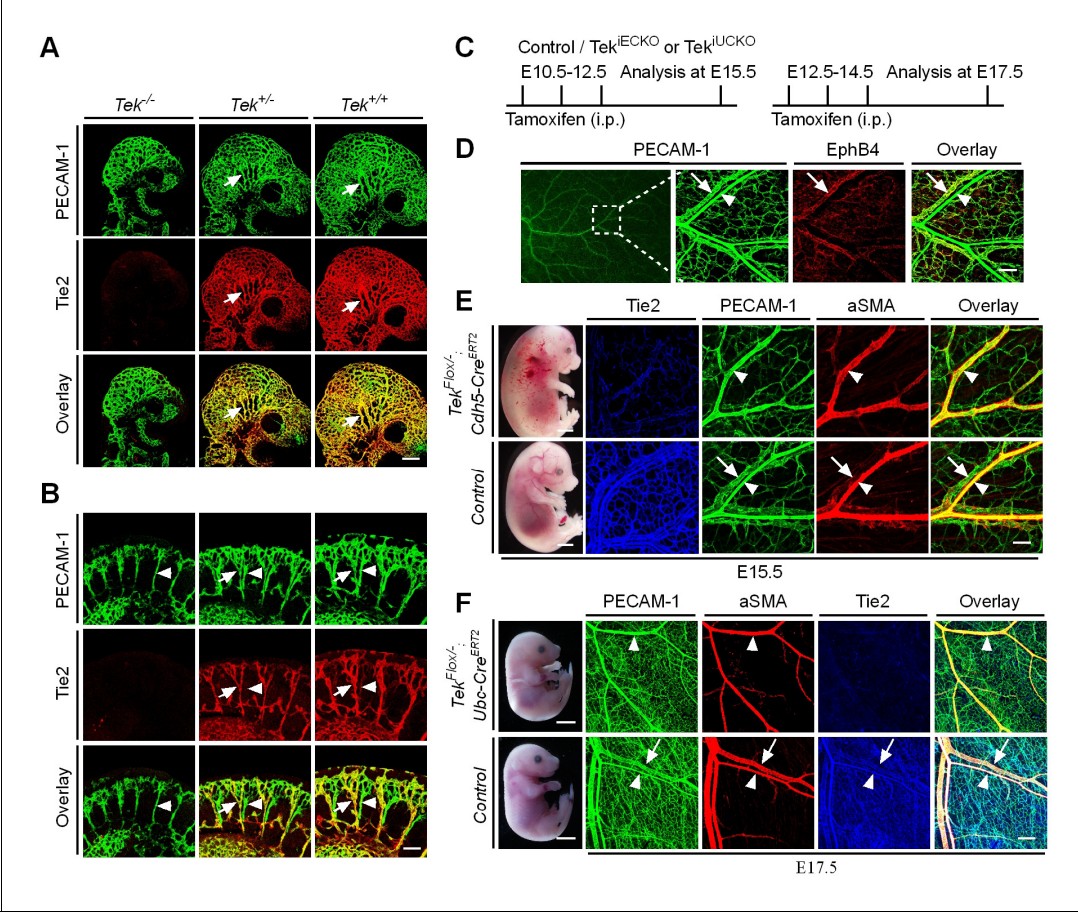

**Figure 1.** Tie2 insufficiency during embryogenesis arrests venous development. (**A**, **B**) Analysis of blood vessels in head (**A**, E9.5) and somites (**B**) by whole-mount immunostaining for PECAM-1 (green) and Tie2 (red). (**C**) Tamoxifen intraperitoneal (i.p.) administration and analysis scheme. (**D–F**) Visualization of veins in E15.5 (**D**, **E**) or E17.5 (**F**) skin of wildtype and *Tek* mutant mice by immunostaining for PECAM-1 (green) and EphB4 or αSMA (red). Arrows point to veins and arrowheads to arteries. The experiments with the ubiquitous or EC-specific *Tek* deletion were repeated for at least three times. Scale bar: 200 μm in **A**, **D** and **F** (4 mm in **F** embryos); 100 μm in **B** and **E** (2 mm in **E** embryos).

The following figure supplements are available for figure 1:

**Figure supplement 1.** Generation and analysis of Tie2 knockout mice.

**Figure supplement 2.** Defective skin vein formation and abnormal arteriovenous alignment in mesenteries of *Tek* mutant mice (Tek$^{-/iUCKO}$).

**Figure supplement 3.** Lymphatic dilation in the skin of mutant mice with Tie2 deletion in vascular endothelial cells.

secondary to tissue edema resulting from the impairment of blood vasculature. However, it is also possible that Tie2 may have a role at earlier stages of lymphatic development.

## Retardation of retinal vascularization after Tie2 attenuation

To study the role of Tie2 in postnatal vascular formation, the $Tek^{Flox/-}/Ubc\text{-}Cre^{ERT2}$ neonatal mice were treated with tamoxifen via intragastric injection during postnatal days (P1–4) and analyzed at P7 (*Figure 2A*). The efficiency of *Tek* deletion was examined by Western blot analysis, immunostaining and real-time RT-PCR. Little Tie2 protein was detected in the lung and retina of Tek$^{-/iUCKO}$ mice (*Figure 2B,C*). The level of Tie2 mRNA in the same tissues of Tek$^{-/iUCKO}$ mice was approximately 13–18% of the control mice as shown in *Table 1*. Consistent with the results from Western blot analysis, the deletion efficiency of Tie2 as analyzed by real-time RT-PCR in lungs of Tek$^{-/iECKO}$ mice was lower than that of Tek$^{-/iUCKO}$ mutants (Tek$^{-/iECKO}$: 0.40 ± 0.10, n = 4; Tek$^{+/iECKO}$: 1.0 ± 0.21, n = 4).

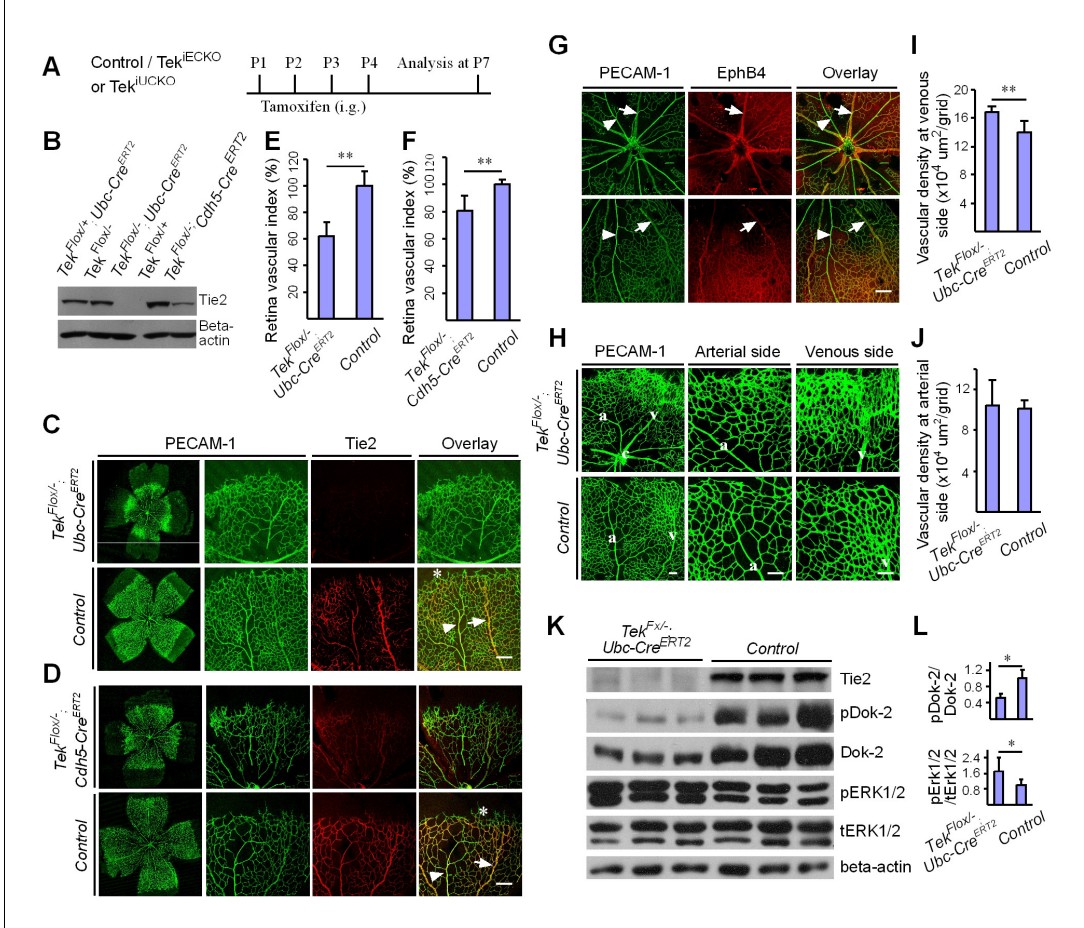

**Figure 2.** Attenuation of Tie2 expression retards retinal vascularization. (**A**) Tamoxifen intragastric (i.g.) administration and analysis scheme. (**B–D**) The deletion efficiency of Tie2 was examined by Western blot analysis (**B**) and immunostaining (**C, D**) for PECAM-1 (green) and Tie2 (red) of tissues from *Tek* deleted and littermate control mice at P7. Note that Tie2 expression is downregulated in tip endothelial cells (asterisk in **C** and **D**) and also low in newly formed retinal arteries (arrowhead in **C** and **D**) when compared with veins (arrow in **C** and **D**). (**E, F**) Quantification of the vascularization index (ratio of vascularized area to total retina area normalized against the littermate controls) after ubiquitous (**E**, Tek$^{-/iUCKO}$: 61.95 ± 10.79, n = 12; Control: 100.0 ± 10.94, n = 12; p<0.0001) or EC-specific deletion (**F**, Tek$^{-/iECKO}$: 80.69 ± 11.13, n = 5; Control: 100.0 ± 3.77, n = 7; p<0.0015). (**G–J**) Visualization of retinal blood vessels (**G, H,** P7) by immunostaining for PECAM-1 (green) and EphB4 (red). Arrows point to veins (v) and arrowheads to arteries (a). Quantification of blood vessel density in the distal retinal venous (**I**, X 10⁴ μm²/grid; Tek$^{-/iUCKO}$: 16.91 ± 0.77, n = 6; Control: 14.07 ± 1.54, n = 8; p=0.0014) and arterial segments (**J**, X 10⁴ μm²/grid; Tek$^{-/iUCKO}$: 10.45 ± 2.47, n = 6; Control: 10.16 ± 0.76, n = 8; p=0.7571) in *Tek* mutants compared with control mice. (**K, L**) Analysis of Dok-2 and ERK1/2 phosphorylation. Total Dok-2, ERK1/2 and beta-actin were used as loading controls. The bands were quantified and normalized against the control group (**L**). Scale bar: 200 μm in **C, D, G**; 100 μm in **H**.

The following figure supplements are available for figure 2:

**Figure supplement 1.** Analysis of Tie2 deletion in Tek$^{-/iECKO}$;mTmG mice.

**Figure supplement 2.** Tie2 deletion in hematopoietic cells does not affect retinal blood vessel growth.

The partial Tie2 deletion was also indicated by the mosaic GFP expression when the mTmG allele was crossed into the *Tek* mutant mice to generate the compound genetic mouse model (Tek$^{-/iECKO}$; *mTmG*; *Figure 2—figure supplement 1A–B*) (*Muzumdar et al., 2007*). Interestingly, the decrease of Tie2 in the Tek$^{-/iUCKO}$ and Tek$^{-/iECKO}$ mice resulted in a significant decrease of retinal vascularization, as indicated by a lower vascular index (*Figure 2C–F*). As Tie2 is also expressed by some hematopoietic cells, we generated mutant mice with *Tek* deletion in blood cells (*Tek$^{Flox/-}$;Vav-iCre*) (*de Boer et al., 2003*). No obvious defects were observed in the retinal blood vessels of the mutant mice (*Figure 2—figure supplement 2A,B*).

**Table 1.** Retina transcript levels of venous and arterial markers.

The mRNA expression level of venous and arterial endothelial cell markers

| Venous and arterial markers | mRNA expression level (retina) | | n (Tek$^{-/UCKO}$) | n (Control) | p value |
|---|---|---|---|---|---|
| | Tek$^{-/UCKO}$ | Control (Tek$^{+/UCKO}$) | | | |
| Tie2 | 0.18 ± 0.09 | 1.0 ± 0.31 | 7 | 7 | <0.0001 |
| APJ | 0.47 ± 0.20 | 1.0 ± 0.36 | 7 | 7 | 0.00488 |
| EphB4 | 0.79 ± 0.12 | 1.0 ± 0.20 | 7 | 7 | 0.0404 |
| COUP-TFII | 0.96 ± 0.11 | 1.00 ± 0.10 | 7 | 7 | 0.449 |
| Dll4 | 0.94 ± 0.16 | 1.0 ± 0.22 | 7 | 7 | 0.565 |
| EphrinB2 | 0.96 ± 0.10 | 1.0 ± 0.24 | 6 | 7 | 0.699 |
| NRP1 | 0.94 ± 0.06 | 1.0 ± 0.24 | 6 | 7 | 0.542 |
| NOTCH1 | 0.98 ± 0.23 | 1.0 ± 0.22 | 6 | 7 | 0.889 |
| Lung transcript levels of venous and arterial markers | | | | | |
| Venous and arterial markers | mRNA expression level (lung) | | n (Tek$^{-/UCKO}$) | n (Control) | p value |
| | Tek$^{-/UCKO}$ | Control (Tek$^{+/UCKO}$) | | | |
| Tie2 | 0.13 ± 0.06 | 1.0 ± 0.29 | 11 | 11 | <0.0001 |
| APJ | 0.55 ± 0.14 | 1.0 ± 0.24 | 11 | 11 | <0.0001 |
| EphB4 | 0.69 ± 0.16 | 1.0 ± 0.16 | 11 | 11 | 0.00026 |
| COUP-TFII | 0.98 ± 0.24 | 1.0 ± 0.14 | 11 | 11 | 0.734 |
| Dll4 | 1.53 ± 0.46 | 1.0 ± 0.19 | 11 | 11 | 0.00207 |
| EphrinB2 | 0.98 ± 0.33 | 1.0 ± 0.23 | 11 | 11 | 0.856 |
| NRP1 | 1.05 ± 0.19 | 1.0 ± 0.26 | 11 | 11 | 0.607 |
| NOTCH1 | 0.93 ± 0.34 | 1.0 ± 0.12 | 11 | 11 | 0.549 |

## Increase of venous angiogenesis in retina after postnatal deletion of Tie2

In spite of the decreased vascular index in retina, there was a significant increase of vascular density at the front of the venous but not arterial segments of Tek$^{-/iUCKO}$ mutants compared with controls at P7 (*Figure 2G–J*). Biochemical analysis revealed a significantly decreased phosphorylation of the Dok-2 docking protein (pDok2 / beta-actin, Tek$^{-/iUCKO}$: 0.38 ± 0.25, n = 6; Control: 1.0 ± 0.24, n = 6; p=0.0014; pDok2 / tDok2, Tek$^{-/iUCKO}$: 0.52 ± 0.10, n = 3; Control: 1.0 ± 0.21, n = 3; p=0.024), but an increase of Erk1/2 phosphorylation (pERK1/2 / tERK1/2, Tek$^{-/iUCKO}$: 1.71 ± 0.70, n = 10; Control: 1.0 ± 0.30, n = 9; p=0.012) in the lungs of Tek$^{-/iUCKO}$ mice compared with controls (*Figure 2K,L*). It is worth noting that the total Dok-2 protein level also decreased in the mutant mice after Tie2 reduction compared with that of the littermate control.

## Tie2 insufficiency leads to vascular tuft formation along retinal veins

The scheme for *Tek* deletion by intragastric administration of tamoxifen was shown in *Figure 3A*. The abnormal angiogenesis along the retinal veins was also seen at P11 in mice with postnatal *Tek* deletion, and the morphology of veins was severely disrupted at P15 (*Figure 3B and C*, arrows). Vein degeneration was accompanied by the formation of haemangioma-like vascular tufts by P21 (*Figure 3D*, arrows). The massive angiogenic vascular growth occurred mainly in the first layer of retinal vessels of Tek$^{-/iUCKO}$ mice (*Figure 3E*). In contrast to wildtype littermate control mice, Tek$^{-/iUCKO}$ mutant mice at P21 exhibited little blood vessel growth towards the deep layers of retina (*Figure 3E*). Vessels in the vascular tufts in the Tek$^{-/iUCKO}$ mice had lumens and were covered with NG2$^{+}$ pericytes (*Figure 3E*). Furthermore, *Tek* deletion at a later stage (P5-8) resulted in a similar but milder vascular phenotype (*Figure 3F*, arrows). For comparison, we also performed the analysis of retinal blood vasculature at different time points with Tek$^{-/iECKO}$ mice, but did not observe the

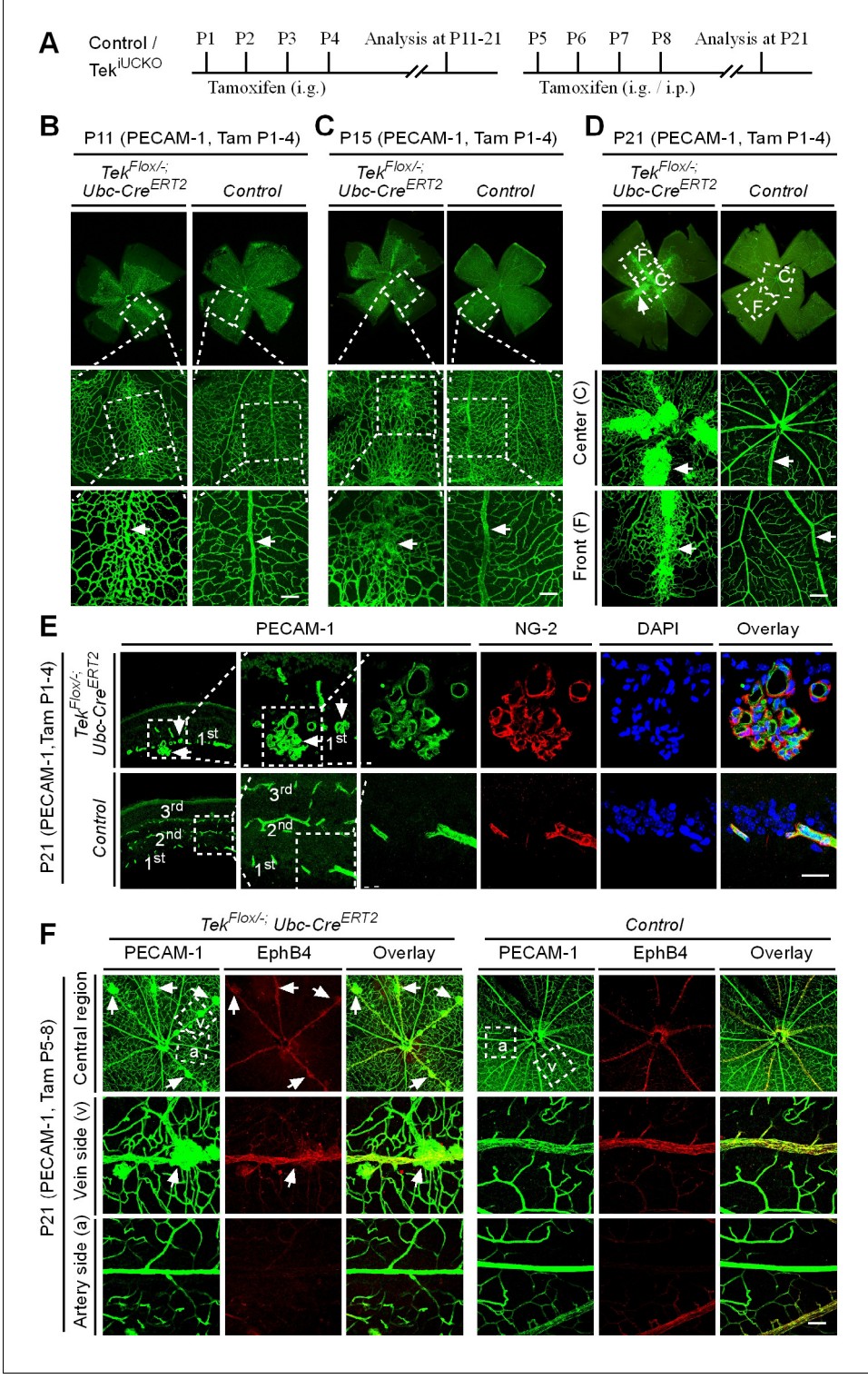

**Figure 3.** Tie2 attenuation leads to vascular tuft formation along retinal veins. (**A**) Tamoxifen intragastric (i.g.) administration and analysis scheme. (**B–D**) Analysis of blood vessels in the retinas of *Tek* deleted (P1–4) and control mice at P11 (**B**), P15 (**C**), and P21 (**D**). (**E**) Cross-sectional analysis of the three layers of retinal blood vessels in *Tek* mutant and control mice. (**F**) Tie2 deletion was induced at P5–8, and analysis of retinal blood vessels was performed at P21. Arrows point to haemangioma-like vascular tufts. The experiments at each time point were repeated for at least three times. Scale bar: 100 μm in **B–D** and **F**, 25 μm in **E**.

*Figure 3 continued*

The following figure supplements are available for figure 3:

**Figure supplement 1.** Growth curves of Tek$^{-/iUCKO}$ mutant and littermate control mice.

**Figure supplement 2.** Analysis of cutaneous blood vessels at P7 and P21, after Tie2 deletion at P1-4.

formation of vascular tufts. This may be due to the lower efficiency of *Tek* gene deletion as discussed above.

Despite the lethality of *Tek* null embryos, the Tek$^{-/iUCKO}$ mice deleted at P1-4 survived with reduced body weight (*Figure 3—figure supplement 1*). Unlike in the retina, postnatal attenuation of Tie2 did not produce an obvious effect on blood vessels in tail skin at P7 (*Figure 3—figure supplement 2A*), or ear skin at P21 (*Figure 3—figure supplement 2B*). However, veins in ear skin were found to be tortuous in 2.5 month-old adult mutants compared with the control mice (*Figure 4A*, arrows). The expression analysis by real-time RT-PCR revealed that *Tek* transcript level remained low in Tek$^{-/iUCKO}$ mice at this stage (Tek$^{-/iUCKO}$: 0.26 ± 0.14, n = 5; Control: 1.0 ± 0.28, n = 5). Interestingly, the venous tortuosity was also observed in retinas of Tek$^{-/iUCKO}$ mice at the adult stage (2.5 month-old) while the increased angiogenesis along the retinal veins regressed (*Figure 4B*). The findings suggest that Tie2 has an important role in the postnatal maintenance of veins.

## Requirement of Tie2 for the maintenance of venous EC identity

Mechanistically, we found that Tie2 attenuation led to the alteration of venous EC identity as shown by the change of EC marker expression (*Figure 5A*, and *Table 1*). Transcript levels of the venous marker APJ and EphB4 were decreased, whereas COUP-TFII mRNA level was unaltered in the retinas and lungs of Tek$^{-/iUCKO}$ mice compared with littermate controls (P7; *Figure 5A*). Furthermore, there was a significant increase of arterial and angiogenic sprout marker Dll4 transcripts in the lungs (*Figure 5A*) (*Hellström et al., 2007*). Consistently, Tie2 reduction induced Dll4 expression in retinal veins (P9; *Figure 5B*, arrows point to veins in dotted regions), which were negative by immunostaining for Dll4 in littermate control mice (*Figure 5C*). There was no significant alteration of other arterial markers including NRP1, EphrinB2 and NOTCH1 (P7; *Figure 5A*). Furthermore, we also examined the expression level of some venous and arterial markers, including EphB4, APJ, Ephrin B2 and Dll4 in *Tek* mutant and control mice at the adult stage. Consistently, we found that venous genes were significantly decreased in lung of Tek$^{-/iUCKO}$ mice compared with controls (EphB4, Tek$^{-/iUCKO}$: 0.52 ± 0.29, n = 5; Control: 1.0 ± 0.18, n = 5; APJ, Tek$^{-/iUCKO}$: 0.22 ± 0.22, n = 5; Control: 1.0 ± 0.25, n = 5). Interestingly, there was also a trend of reduction in the arterial gene expression in Tek$^{-/iUCKO}$ mice (Ephrin B2, Tek$^{-/iUCKO}$: 0.71 ± 0.32, n = 5; Control: 1.0 ± 0.23, n = 5; Dll4, Tek$^{-/iUCKO}$: 0.70 ± 0.36, n = 5; Control: 1.0 ± 0.21, n = 5), suggesting that Tie2 retardation may also affect arteries at later stages.

## Regulation of COUP-TFII protein stability by Tie2/Akt pathway

Tie2 deletion led to a decrease of Akt phosphorylation (Ser473, *Figure 6A,B*). Interestingly, COUP-TFII protein was significantly decreased in lung and liver tissues of *Tek* mutant mice (*Figure 6A,B* and *Figure 6—figure supplement 1A,B*), although COUP-TFII mRNA level was not altered, as shown above. This was further verified in cultured HUVECs, where COUP-TFII protein was decreased when Tie2 was reduced by siRNA mediated knockdown (*Figure 6C–D*). Consistently, COUP-TFII protein was significantly increased by the COMP-Ang1 stimulation of Tie2/Akt pathway (*Figure 6E–F*). As the Tie2 downstream Akt activation was significantly suppressed in mice with Tie2 attenuation, we also analyzed COUP-TFII level when the PI3K/Akt pathway was blocked. COUP-TFII protein was significantly reduced 6 hr or 12 hr after treatment with the PI3K inhibitor LY294002 (*Figure 6G,H*; LY-3h: 1.11 ± 0.39, n = 5; LY-6h: 0.62 ± 0.12, n = 5; LY-12h: 0.39 ± 0.13, n = 5; values from five independent experiments normalized against control at three time points respectively). This was further confirmed with the Akt inhibitor MK2206, and there was a significant decrease of COUP-TFII after Akt inhibition at the 12 hr time point (*Figure 6I,J*; MK-3h: 1.32 ± 0.47, n = 7; MK-6h: 0.75 ± 0.31, n = 7; MK-12h: 0.23 ± 0.18, n = 7; values from seven independent experiments normalized against

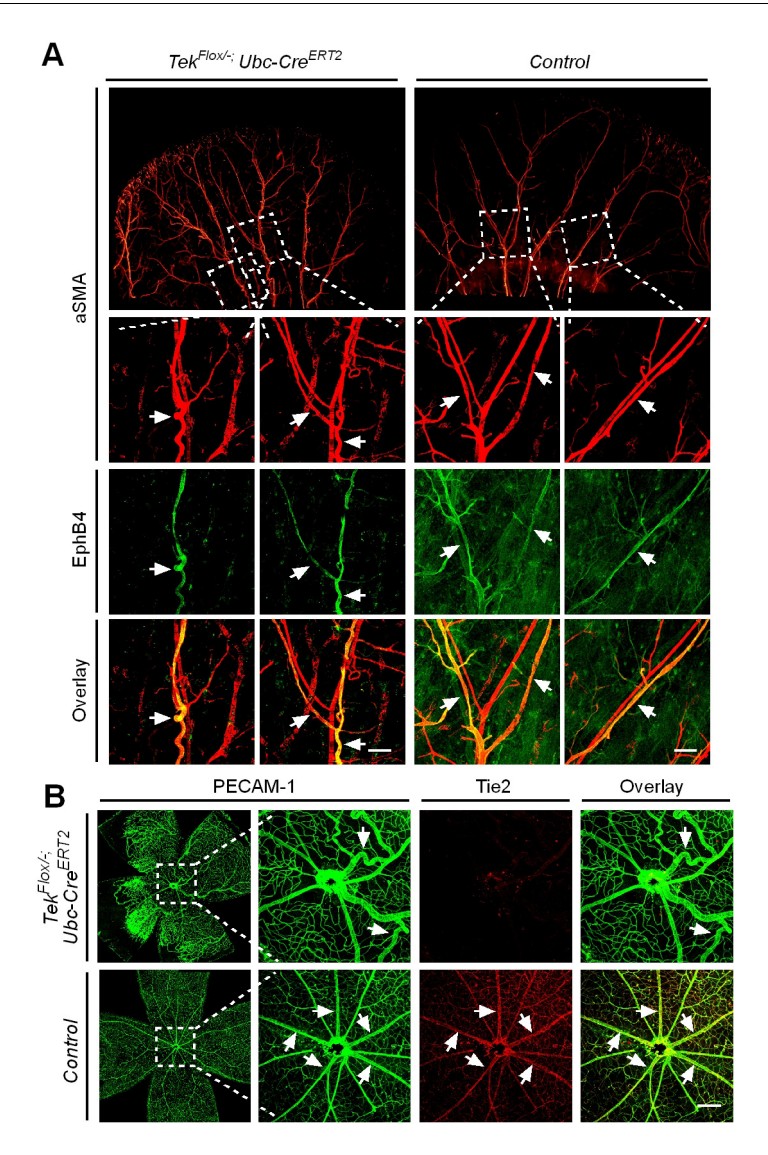

**Figure 4.** Effect of long-term postnatal attenuation of Tie2 on cutaneous and retinal veins. (**A**) Analysis of blood vessels in the ear skin of *Tek* mutant and control mice by immunostaining for αSMA (red) and EphB4 (green). (**B**) Visualization of retinal blood vessels by immunostaining for Tie2 (red) and PECAM-1 (green). Note tortuous veins in ear skin and retinas in 2.5 month old Tek$^{-/iUCKO}$ mice after Tie2 deletion at P1–4. Arrows point to veins. The histological analysis for both groups was repeated for at least three times. Scale bar: 200 μm in **A**, **B**.

control at three time points respectively). Furthermore, the decrease of COUP-TFII protein by the Akt inhibition could be reverted by the treatment with a proteasome inhibitor MG132 (*Figure 6K–L*). These findings suggest that the Tie2 signaling pathway controls vein specification via Akt-mediated regulation of COUP-TFII protein stability (*Figure 6M*).

## Discussion

We show here that Tie2 is more expressed in veins than arteries in early embryos and in retina of neonate mice with the newly formed arteries expressing low level of Tie2. Interestingly, Tie2 is also downregulated in the sprouting tip endothelial cells as observed in retinal angiogenesis in this study and also by others (*Augustin et al., 2009*). It has been recently reported that vein-derived

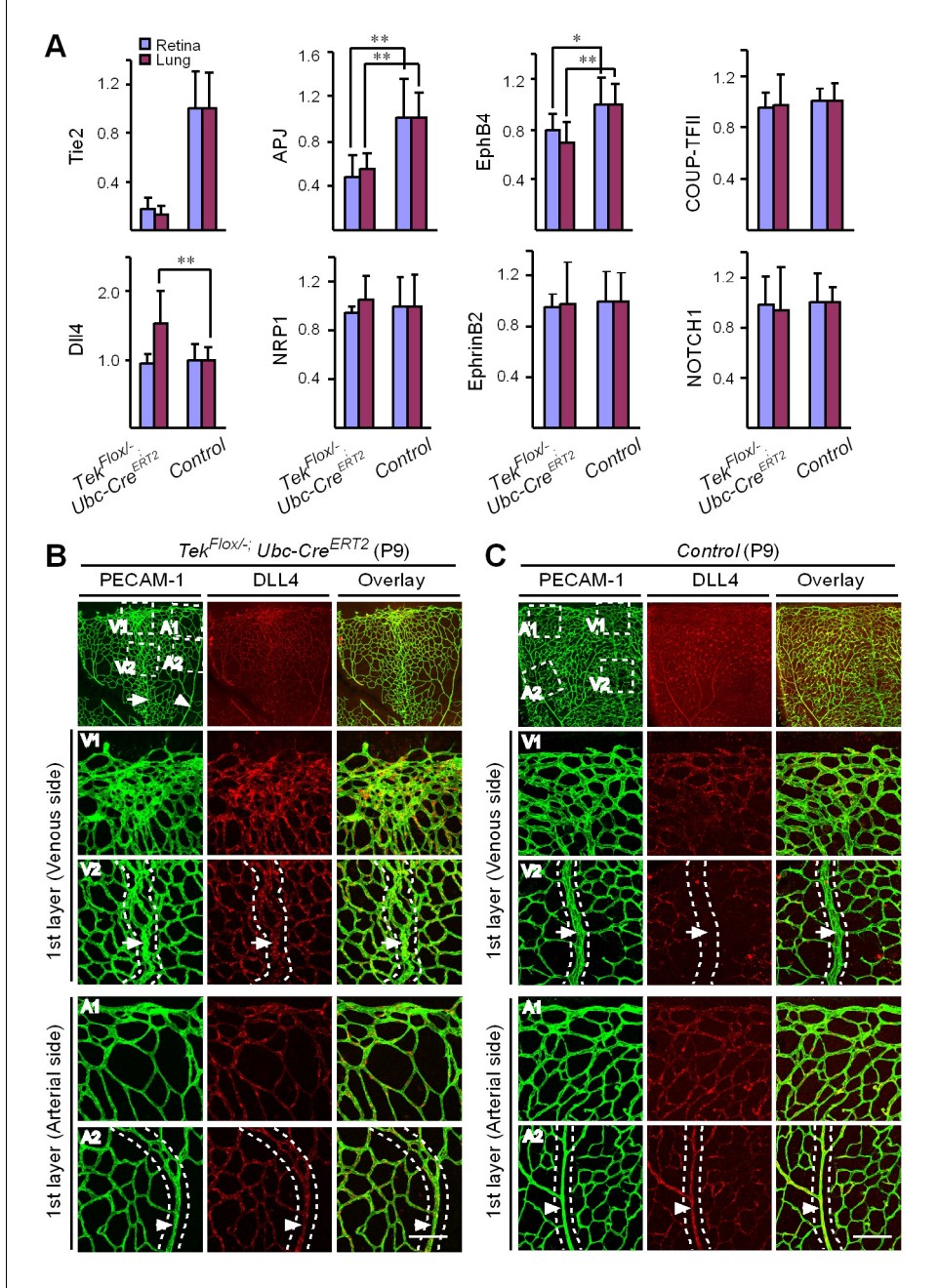

**Figure 5.** Requirement of Tie2 for the maintenance of venous EC identity. (**A**) Quantitative expression analysis of venous markers including EphB4, APJ and COUP-TFII, and arterial markers including EphrinB2, NRP1, NOTCH1 and Dll4 in lung and retina tissues of *Tek* mutant and control mice (P7). (**B**, **C**) Immunostaining for Dll4 with retinas of *Tek* mutant and control mice (P9). Scale bar: 100 μm in **B** and **C**.

endothelial tip cells contribute to the emerging arteries during mouse retinal vascularization and also in zebrafish fin regeneration (*Xu et al., 2014*). This suggests that the downregulation of Tie2 may be required for the initial establishment of an arterial EC identity. Consistently, we have demonstrated in this study that Tie2 absence or insufficiency by gene targeting disrupts venogenesis during embryogenesis and postnatal development. At the molecular level, we have found that Tie2 participates in the determination of venous EC identity, which may act via Akt-mediated regulation of COUP-TFII protein stability. This implies that Tie2/Akt signaling counterbalances VEGFR2/MAPK

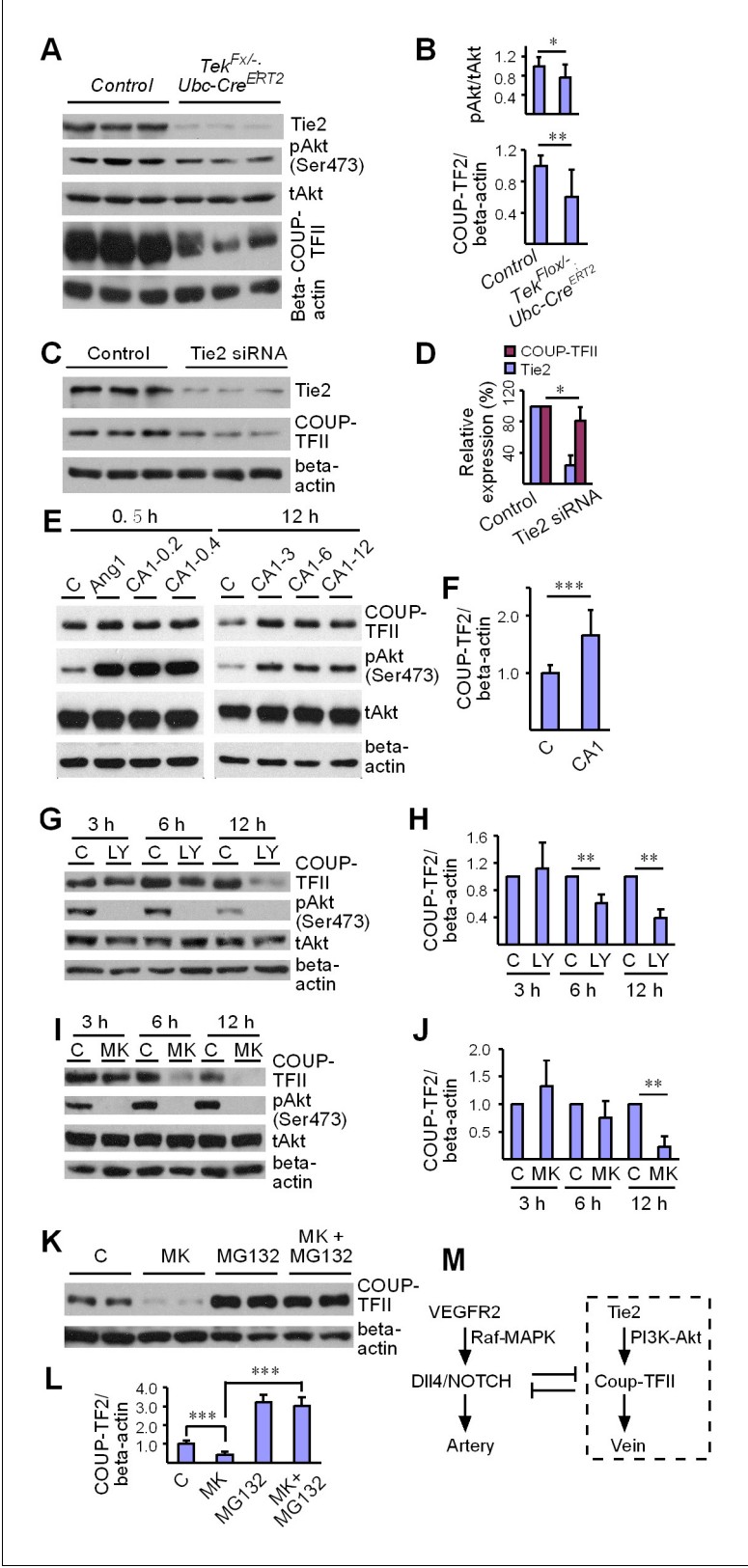

**Figure 6.** Regulation of COUP-TFII protein level by Tie2 pathway. (**A, B**) Analysis and quantification of Akt (Ser473) phosphorylation (pAkt / tAkt, Tek$^{-/iUCKO}$: 0.76 ± 0.27, n = 10; Control: 1.0 ± 0.19, n = 9; p<0.0407) and COUP-TFII protein in lungs of *Tek* mutant and control mice, and total Akt and beta-actin as loading controls (COUP-TFII/

*Figure 6 continued on next page*

*Figure 6 continued*

beta-actin; Tek$^{-/iUCKO}$: 0.60 ± 0.35, n = 12; control: 1.0 ± 0.13, n = 12; p<0.01). (C, D) Western blotting of COUP-TFII protein in HUVECs after siRNA mediated Tie2 knockdown. COUP-TFII protein level decreased to 80.55 ± 17.79% of the control when Tie2 expression was reduced to 24.33 ± 13.05% of control (normalized by beta-actin; values from six independent experiments). (E, F) Increase of COUP-TFII (COUP-TFII/beta-actin; COMP-Ang1: 1.66 ± 0.45, n = 12; Control: 1.0 ± 0.15, n = 10; p<0.001; values from five independent experiments) by COMP-Ang1 activation of the Tie2/Akt pathway for 12 hr with the recombinant protein added every 4 hr (CA1–3, 0.2–0.4 μg/ml). Note that COMP-Ang1 added every 1 hr (CA1–12) or 2 hr (CA1–6) did not further increase the level of COUP-TFII, and that treatment with Ang1 or COMP-Ang1 for half an hour did not produce an obvious difference in the COUP-TFII protein level. (G–J) Analysis of COUP-TFII in HUVECs after treatment with LY294002 or MK2206 to inhibit the PI3K/Akt signaling pathway for 3 hr, 6 hr or 12 hr. (K, L) Decrease of COUP-TFII by the Akt inhibitor MK2206 was blocked by the proteasome inhibitor MG132 when analyzed 12 hr after the treatment (COUP-TFII/beta-actin; Control: 1.0 ± 0.16, n = 5; MK2206: 0.42 ± 0.17, n = 5; MG132: 3.20 ± 0.38, n = 7; MK2206 + MG132: 3.01 ± 0.46, n = 7; p<0.001; values from three independent experiments). (M) Schematic model of Tie2 mediated signaling in vein specification via the Akt mediated regulation of COUP-TFII. The Tie2/Akt pathway may counterbalance VEGFR2/MAPK signaling during arteriovenous specification.

The following figure supplement is available for figure 6:

**Figure supplement 1.** Analysis of COUP-TFII in the liver.

---

pathway in arteriovenous specification during the vascular development. The findings are consistent with previous literature on the role of Akt in venous development, and with the recent report that cardiomyocyte derived Ang1 is required for the subepicardial coronary vein formation (*Deng et al., 2013*; *Ren et al., 2010*; *Zimmermann and Moelling, 1999*; *Lamont and Childs, 2006*; *Arita et al., 2014*).

As Tie2 is expressed by endothelial cells and non-endothelial cells such as hematopoietic cells, we employed three Cre deletors in this study to analyze the role of Tie2 in blood vascular development, including a EC-specific Cre line *Cdh5-Cre$^{ERT2}$*, a hematopoietic Cre line *Vav-iCre* and a ubiquitous Cre line *Ubc-Cre$^{ERT2}$*. Blood vessels developed normally in mice with *Tek* deletion in blood cells, while disruption of vein development was observed in Tek$^{-/iUCKO}$ as well as in Tek$^{-/iECKO}$ mutant mice. This suggests that the disruption of cutaneous vein development during embryogenesis is likely to be a cell-autonomous effect resulting from the deletion of Tie2 in endothelial cells. In postnatal studies, we found that there was a dramatic increase of blood vessel growth after the ubiquitous Tie2 attenuation, which led to the formation of haemangioma-like vascular tufts along the retinal veins in the *Tek* mutant mice. Although a significant decrease of retinal vascularization was observed in Tek$^{-/iECKO}$ mice, we did not observe the vascular tuft formation in the retina of these mutants. This may result from the lower Tie2 deletion efficiency in Tek$^{-/iECKO}$ mice when compared with that of Tek$^{-/iUCKO}$ mice. However, we cannot completely rule out the possibility that Tie2 deletion in other non-endothelial cells may contribute to the vascular abnormalities in retinas of Tek$^{-/iUCKO}$ mutant mice. As retinal neovascularization is one of the frequent causes of vision loss in patients with proliferative diabetic retinopathy and neovascular age-related macular degeneration (*Gariano and Gardner, 2005*), these data suggests that reduction of Tie2 mediated signals in ECs and / or other Tie2 expressing cells may be implicated in the vascular pathologies.

COUP-TFII is expressed by several cell types including endothelial cells (*Wu et al., 2016*). In the vascular system, COUP-TFII has been shown to regulate venous EC identity as well as lymphatic EC specification (*You et al., 2005*; *Lin et al., 2010*; *Srinivasan et al., 2010*). At the transcription level, COUP-TFII is regulated by several factors / pathways including NOTCH and SOX7/18 (*Swift et al., 2014*). In this study, we have found that COUP-TFII protein but not its mRNA level is significantly reduced in genetically engineered mouse model targeting *Tek*. Based on the evidence obtained from the biochemical analysis with cultured endothelial cells, we propose that Akt is a strong candidate mediating Tie2 signals in this process. As Akt acts downstream of many signaling pathways and that it is also activated in many cell types, it remains to be investigated how Akt regulates COUP-TFII protein stability in venous ECs. Also, further in vivo mechanistic studies are required for validating and fully elucidating Akt and others factors / pathways downstream of Tie2 in venous specification. In addition, we have also found that the requirement of Tie2 for vein development differs

between the skin and mesentery. When Tie2 levels were reduced by gene targeting, veins were mostly undetectable in the skin, but displayed only abnormal arterial-venous alignment in the mesentery. It is possible that the cutaneous veins are more sensitive to the loss of Tie2 mediated signals, or that venogenesis occurs earlier in the mesentery than in the skin. The abnormal arteriovenous alignment resembled that of the APJ deficient mice (*Kidoya et al., 2015*), and a significant reduction of APJ in the *Tek* mutants suggests that APJ acts downstream of Tie2 in the regulation of venous patterning.

Interestingly, we have also found that both the total and phosphorylated Dok-2 decreased significantly after Tie2 attenuation in this study. As Dok family members have been shown to negatively regulate ERK1/2 activation (*Honma et al., 2006*), the decrease of Dok-2 may lead to the increase of retinal angiogenesis as detected in mice with Tie2 reduction by gene targeting. Consistently, enhancement of Tie2 signaling via the inhibition of VE-PTP has been shown to stabilize blood vessels and suppress retinal angiogenesis in mouse models of ischemia-induced retinal neovascularization (*Shen et al., 2014b*). Furthermore, in experiments using HEK 293 T or endothelial cells overexpressing Dok2, it was found that Dok2, via directly interacting with Tie2, mediated Tie2 pathway in endothelial cell migration (*Saharinen et al., 2008*; *Master et al., 2001*). Decreased Dok-2 activation in *Tek* mutants may also account for the suppression of horizontal and vertical blood vessel growth during retinal vascular network formation. So far, in vivo evidence about the role of Dok2 in blood vascular development is still missing. Further studies employing animal models targeting *Dok2* are important to analyze its biological functions in the vascular system.

In summary, we show that Tie2 is required for the specification and maintenance of venous EC identity. Related to the findings of this study, activating mutations of Tie2 have been linked to venous malformations in patients (*Vikkula et al., 1996*). It is still poorly understood about the difference between the pathways downstream of activating Tie2 mutants and the wildtype Tie2. Activation of wildtype Tie2 by its ligand Ang1 or VE-PTP inhibition, at least for a short period, has been shown to stabilize blood vasculature (*Shen et al., 2014b*), while activating Tie2 mutations lead to venous malformations (*Vikkula et al., 1996*; *Limaye et al., 2009*; *Boscolo et al., 2015*). Mechanisms underlying the discrepancy require further investigation. As alteration of Tie2 mediated signals may be implicated in a variety of vascular pathologies associated with the venous system, further investigation along these lines may help to develop novel Tie2-targeted therapeutics.

## Materials and methods

### Animal models

Conditional mice with *Tek* gene targeted flox sites (*Tek^Flox^*) for gene deletion were generated by the National Resource Center for Mutant Mice, Nanjing University, as previously described (*Shen et al., 2014a*). All animal experiments were performed in accordance with the institutional guidelines of the Soochow and Nanjing University Animal Center (MARC-AP#YH2). To generate mice with ubiquitous or cell-specific *Tek* gene deletion, *Tek^Flox^* mice were crossed with transgenic mice expressing Cre recombinanse in ECs (*Cdh5-Cre^ERT2^*) (*Wang et al., 2010*), hematopoietic cells (*Vav1-iCre*) (*de Boer et al., 2003*), or ubiquitously (*Ubc-Cre^ERT2^* or *EIIa-Cre*) (*Lakso et al., 1996*; *Ruzankina et al., 2007*). The floxed mice used in this study were maintained in C57BL/6J (RRID: IMSR_JAX:000664) with at least five backcrosses. In all the phenotype analysis, littermates were used as control.

### Induction of gene deletion

Induction of gene deletion was performed by tamoxifen treatment as previously described (*Shen et al., 2014a*). Briefly, pregnant mice were treated with tamoxifen (Sigma-Aldrich) at E10.5–12.5 or E12.5–14.5 (1–2 mg/per mouse for three consecutive days by intraperitoneal injection), and analyzed later. New-born pups were treated with tamoxifen by four daily intragastric injections and after P7 by four daily intraperitoneal injections, and analyzed later.

### Quantitative real-time RT-PCR

Tissues from *Tek* mutant and control mice were collected and homogenized in TRIzol (Ambion). RNA extraction and reverse transcription were performed by standard procedures (RevertAid First Strand

cDNA Synthesis Kit, Thermo Scientific). Quantitative real-time RT–PCR was carried out using the SYBR premix Ex Taq kit (TaKaRa). Briefly, for each reaction, 50 ng of total RNA was transcribed for 2 min at 50°C with a denaturing step at 95°C for 30 s followed by 40 cycles of 5 s at 95°C and 34 s at 60°C. Fluorescence signal was analyzed by using ABI PRISM 7500. The primers used were as follows: GAPDH: 5'-GGTGAAGGTCGGTGTGAACG-3', 5'-CTCGCTCCTGGAAGATGGTG-3'; Tie2: 5'-GATTTTGGATTGTCCCGAGGTCAAG-3', 5'-CACCAATATCTGGGCAAATGATGG-3'; APJ: 5'-CAGTCTGAATGCGACTACGC-3', 5'-CCATGACAGGCACAGCTAGA-3'; Ephb4: 5'-CTGGATGGAGAACCCCTACA-3', 5'-CCAGGTAGAAGCCAGCTTTG-3'; COUP-TFII: 5'-GCAAGTGGAGAAGCTCAAGG-3', 5'-TTCCAAAGCACACTGGGACT-3'; NRP1: 5'-CCGGAACCCTACCAGAGAAT-3', 5'-AAGGTGCAATCTTCCCACAG-3'; EphrinB2: 5'-TGTTGGGGACTTTTGATGGT-3', 5'-GTCCACTTTGGGGCAAATAA-3'; NOTCH1: 5'-TGTTGTGCTCCTGAAGAACG-3', 5'-TCCATGTGATCCGTGATGTC-3'; Dll4: 5'-TGCCTGGGAAGTATCCTCAC-3', 5'-GTGGCAATCACACACTCGTT-3'. The transcripts of venous and arterial markers were normalized against GAPDH, and the relative expression level of every gene in the *Tek* mutants (Tek$^{-/iUCKO}$) was normalized against that of littermate control mice.

## Western blot analysis

To analyze Tie2 mediated signaling pathway, lung tissues from *Tek* mutant and control mice were homogenized following standard procedures. Antibodies used included rabbit polyclonal anti-Tie2 (Santa Cruz sc-324, RRID:AB_631102), rabbit polyclonal anti-Akt (Cell Signaling Technology #9272, RRID:AB_329827), rabbit monoclonal anti-phospho-Akt473 (Cell Signaling Technology, #4060), rabbit polyclonal anti-phospho-p56Dok-2 (Cell Signaling Technology #3911, RRID:AB_2095082), rabbit polyclonal anti-p56Dok-2 (Cell Signaling Technology #3914, RRID:AB_2095080), rabbit polyclonal anti-Erk1/2 and phospho-Erk1/2 (Cell Signaling Technology #9101, RRID:AB_331646; and 9102, AB_330744), mouse monoclonal anti-COUP-TFII (R and D Systems #PP-H7147-00, RRID:AB_2155627), and mouse monoclonal to beta–actin antibody (C4, Santa Cruz sc-47778, RRID:AB_626632).

## Cell culture, siRNA transfection and Akt inhibition

HUVECs (human umbilical vein endothelial cell, C0035C, GIBCO) were cultured in endothelial cell basal medium plus supplements (#M200-500 GIBCO, or #1001 ScienCell Research Laboratories). To knock down Tie2 expressoin in HUVECs, cells were transfected with Stealth RNAi™ siRNA duplex oligoribonucleotides targeting human Tie2 (HSS110623, HSS110624 and HSS110625; Invitrogen) using Lipofectamine RNAiMax (Invitrogen, CA, USA); Stealth RNAi negative control duplexes (medium GC) were used as a control. COMP-Ang1 (kind gift from Dr. Gou Young Koh; 0.2–0.4 μg/ml) or Ang1 (923-AN, R and D) was used to stimulate Tie2/Akt pathway. To inhibit PI3K and downstream Akt signaling, HUVECs were treated with LY294002 (40 μM, S1105, Selleckchem) or MK2206 (10 μM, S1078, Selleckchem) and analyzed at 3 hr, 6 hr or 12 hr after treatment. For the experiment with proteasome inhibition, HUVECs were treated with MG132 (10 μM, s2619, Selleckchem). The cells were washed with ice-cold PBS and lysed in the lysis buffer (1 mM PMSF, 2 mM Na$_3$VO$_4$, 1× protease and phosphatase inhibitor cocktail without EDTA (Roche Applied Science), 20 mM Tris-HCl pH 8.0, 100 mM NaCl, 10% glycerol, 50 mM NaF, 10 mM $\beta$-glycerolphosphate, 5 mM sodium pyrophosphate, 5 mM EDTA, 0.5 mM EGTA and 1% NP-40). The lysates were incubated on ice for 0.5 hr with rotation and centrifuged. Protein concentration was determined using the BCA protein assay kit (PIERCE), and equal amounts of protein were used for analysis.

## Immunostaining

For whole-mount immunostaining with embryonic skin, mesentery and retina, tissues were harvested and processed as previously described (*Shen et al., 2014a*). The tissues were fixed in 4% paraformaldehyde, blocked with 3% (w/v) milk in PBS-TX (0.3% Triton X-100), and incubated with primary antibodies overnight at 4°C. The antibodies used were rat anti-mouse PECAM-1 (BD Pharmigen, 553370, RRID:AB_394816), goat anti-mouse Tie2 (R and D, AF762, RRID:AB_2203220), goat anti-mouse Dll4 (R and D, AF1389, RRID:AB_354770), goat anti-mouse EphB4 (R and D, AF446, RRID:AB_2100105), Cy3-conjugated mouse anti-mouse αSMA (Sigma, C6198, RRID:AB_476856). Alexa488, Alexa594 (Invitrogen), Cy5- or Cy3- (Jackson) conjugated secondary antibodies were used for staining. Slides were mounted with Vectashield (VectorLabs), and analyzed with the Olympus FluoView 1000 confocal microscope or Olympus BX51 fluorescent dissection microscope. For

staining of frozen sections, retinas were collected and fixed in 4% PFA for 1 hr at 4°C, incubated in 20% sucrose overnight and then embedded in OCT. Consecutive sections (10 μm in thickness) were incubated with antibodies against PECAM-1 (BD Pharmingen, 553370, RRID:AB_394816) and NG2 (Millipore, AB5320, RRID:AB_11213678), followed by staining with the appropriate fluorochrome-conjugated secondary antibodies and mounted as described above.

### Analysis of retinal vascularization

Retinal vascularization index was quantified as the ratio of vascularized area to total retinal area. For the quantification of blood vessel parameters in the retina, fluorescent images were taken from similar regions in all samples. Blood vessel density was measured and analyzed by using Image Pro Plus (MediaCybernetics), as previously described (*Shen et al., 2014a*).

### Statistical analysis

Statistical analysis was performed with the unpaired *t* test. All statistical tests were two-sided. Data are presented as mean ± S.D.

## Acknowledgements

We thank Dr Dietmar Vestweber and Martina Dierkes for the discussion and kind help with experiments using COMP-Ang1, and staff in the Animal facility of Soochow University and Model Animal Research Institute of Nanjing University for technical assistance. This work was supported by grants from the National Natural Science Foundation of China (91539101, 31271530, 31071263), the Ministry of Science and Technology of China (2012CB947600), and the Priority Academic Program Development of Jiangsu Higher Education Institutions.

## Additional information

### Competing interests

KA: Reviewing editor, *eLife*. The other authors declare that no competing interests exist.

### Funding

| Funder | Grant reference number | Author |
|---|---|---|
| National Natural Science Foundation of China | 91539101 | Yulong He |
| National Natural Science Foundation of China | 31271530 | Yulong He |
| National Natural Science Foundation of China | 31071263 | Yulong He |
| Ministry of Science and Technology of the People's Republic of China | 2012CB947600 | Yulong He |
| The priority of Academic Program Development of Jiangsu Higher Education Institutions | | Yulong He |

The funders had no role in study design, data collection and interpretation, or the decision to submit the work for publication.

### Author ORCIDs

Fei Zhou, http://orcid.org/0000-0003-1857-8831
Ying Xu, http://orcid.org/0000-0002-6689-7768
Yulong He, http://orcid.org/0000-0002-0099-3749

### Ethics

Animal experimentation: Conditional mice with Tek gene targeted flox sites for gene deletion were generated by the National Resource Center for Mutant Mice, Nanjing University. All animal

experiments were performed in accordance with the institutional guidelines of the Soochow and Nanjing University Animal Center (MARC-AP#YH2).

## Author contributions
MC, XC, Acquisition of data, Analysis and interpretation of data, Drafting or revising the article ; TL, BS, HZ, LZ, FZ, WM, PX, ZL, Acquisition of data, Analysis and interpretation of data, Drafting or revising the article; HJ, Acquisition of data, Analysis and interpretation of data, Drafting or revising the article; ND, YX, YZ, GX, QW, GYK, Analysis and interpretation of data, Drafting or revising the article; PL, KA, RHA, Analysis and interpretation of data, Drafting or revising the article ; JL, Analysis and interpretation of data, Drafting or revising the article; YH, Conception and design, Acquisition of data, Analysis and interpretation of data, Drafting or revising the article

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
