## [Decision Letter]

[Editors’ note: a previous version of this study was rejected after peer review, but the authors submitted for reconsideration. The first decision letter after peer review is shown below.]

Thank you for submitting your work entitled "Angiopoietin receptor Tie2 is required for vein specification and maintenance via regulating COUP-TFII" for consideration by *eLife*. Your article has been reviewed by three peer reviewers, one of whom is a member of our Board of Reviewing Editors, and the evaluation has been overseen by Janet Rossant as the Senior Editor. The following individuals involved in review of your submission have agreed to reveal their identity: Hellmut Augustin (Reviewer #2); Lena Claesson-Welsh (Reviewer #3).

Our decision has been reached after consultation between the reviewers. Based on these discussions and the individual reviews below, we regret to inform you that your work will not be considered further for publication in *eLife*.

All the reviewers agree that the paper contains interesting and rather novel observations. However they feel that the paper needs additional experiments to make the results stronger and to get a more mechanistic view of the effect of Tie2 in venous differentiation. Both reviewers 2 and 3 list major points to be considered and specific experiments to add to improve the manuscript. Since these revisions require considerable additional work, we are not able to consider a revised manuscript.

We encourage the authors to complete the recommended revision of their work and consider resubmitting to *eLife* in the future.

Reviewer #1:

The paper by Man Chu et al. describes the effect of inactivation of Tie-2 expression on artero-venous differentiation in the mouse embryo, pups and cultured endothelial cells.

As previously reported, Tie 2 null embryos die early during gestation for vascular and heart developmental problems. Here the authors show that Tie-2 deficient embryos and pups present a defective venous organization when Tie-2 expression is missing. In some organs and tissues, veins are absent along the arteries. In the retina there is a reduced horizontal growth of the vasculature and the presence of malformations along the length of the veins. In the skin, veins look tortuous. Arterial and venous markers are partially modified. In HUVEC silenced for Tie2 or lung and liver tissues Coup-TFII is reduced.

Mechanistically, data suggest that Tie-2 may act through phosphorylation of Akt that in turn would increase COUP-TFII and venous differentiation.

Although the paper contains some interesting observations it is essentially descriptive. The major weakness is the lack of a detailed mechanistic analysis of how Tie-2 acts in inducing venous differentiation. Data on the role of Akt are suggestive but a marked reduction of COUP-TFII is detectable only in HUVEC KD for Tie-2 and is much less apparent in the tissues. Other published studies showed an increase in cultured cells of Akt and STAT 1 upon infection with Tie-2 gain of function mutant. In addition, inherited gain of function mutations of Tie-2 induce a strong venous endothelial cell proliferation and formation of large hemangiomas inhibited by Rapamycin (for instance Boscolo et al. J Clin invest). To my view the authors should extend their studies on the mechanism of action of Tie-2 also adding novel observations as compared to previously published literature.

Reviewer #2:

The authors of the manuscript, "Angiopoietin receptor Tie2 is required for vein specification and maintenance via regulating COUP-TFII" have used global and endothelial cell-specific conditional Tie2 knockout mouse models to demonstrate the critical role of Tie2 during vein development. The study is novel and addresses a fundamental question regarding the pathways (Tie2, Akt/PI3K and COUP-TFII) involved in forming venous endothelial identity and function. Although there is already a lot of circumstantial evidence in the published literature on vein-specific functions of Ang/Tie signaling, the manuscript makes an important point in showing this conclusively using definite genetic models. The study will be of interest for the broad readership of the journal. In further advancing their work, the authors should consider the following:

1) The major limitation in the study is the use of a ubiquitously expressed Cre deletor line (*UBC-Cre^ERT2^*) to knockout Tie2. Even though the authors have also used an endothelial specific Cre line (*Cad5-Cre^ERT2^*) to demonstrate the phenotypes in embryos and early postnatal retina, all the mechanistic studies are only performed using the *UBC-Cre^ERT2^* line. This is a major concern especially because *Tie2^-/iUCKO^* seems to show better Tie2 deletion as well as stronger phenotypes compared to *Tie2^-/iECKO^* line. Is this solely due to different recombination efficacies of the different drivers or would this difference raise the question whether endothelial cell autonomous effects may not be the only cause of defective vein development in Tie2-deficient mice? Why did the authors opt to study different time points in Figure 1?

2) The immunofluorescence images show essentially black boxes in Figure 1 and in Figure 2 suggesting 100% deletion efficacy. Do the authors claim to have achieved 100% deletion of Tie2 in these experiments or why are these images not showing the expected mosaic staining pattern resulting from partial recombination? A quantitative analysis of the efficacy of Tie2 deletion in the different experiments should/needs to be included. Alternatively, qRT-PCR to show Tie2 expression should be done with retina and embryo tissues from *Tie2^-/iUCKO^* (tissues and isolated endothelial cells). In Figure 2, the WB shows about 30-50% Tie2 in the *Tie2^-/iECKO^* mice which is clearly not corresponding to the extremely weak or absent Tie2 staining shown in Figure 2.

3) The retina vasculature analysis at P11, P15 and P21 in *Tie2^-/iUCKO^* mice showing vascular tuft formation is interesting. The same analysis should be included using the *Tie2^-/iECKO^* mice.

4) Although hemangioma-like vascular tufts were observed in *Tie2^-/iUCKO^* mice at P21, the adult retina did not show any such malformations, whereas ear skin showed tortuous veins in adult mice. These data have been discussed rather insufficiently. The authors should comparatively analyze COUP-TFII transcript levels and other downstream players such as pAkt and pDok-2 in these two vascular beds in order to dissect the mechanisms, which help the retinal veins to "recover" but not the cutaneous veins. Likewise, partial recombination will lead to a mosaic vasculature of WT and KO cells. If such mosaic mice are traced over time, there is a good chance that the KO cells will be competed out by WT cells. As such, it is critical, particularly in the longer term experiments to carefully trace the percentage of cells with Tie2 deletion at the later time points of analysis.

5) The authors claim that Tie2 is required for the maintenance of venous EC identity. This is suggested by the observed phenotypes (albeit only moderately convincing). A systematic gene expression analysis of adult retina similar to Figure 5 (performed using P7 retina and lung) is missing. It is important to know whether the expression of venous markers is restored over time and also whether Tie2 expression is recovered in the adult mice as a result of incomplete Cre recombination efficiency and compensatory proliferation of Tie2-expressing EC (see above).

6) The authors are encouraged to reconsider their discussion. They conclude at the end of the discussion that the findings put a cautionary note on long term Tie2 targeting therapies, e.g., during tumor angiogenesis. This may not be the most important point of the study. In fact, nobody tries to target Tie2. Ang2 neutralizing strategies primarily work cooperatively with VEGF pathway targeting drugs based on their vascular normalization Tie2 gain-of-function effect. The same holds true for VE-PTP inhibitors or the recently published ABTAA antibody. What this reviewer finds much more exciting is the role of Tie2 on venogenesis. Venogenesis has mostly been considered as the default pathway with transcriptional programs driving lymphatic and arterial differentiation. The identification of Tie2 signaling as upstream regulator of COUP-TFII is a major finding that should likely be discussed in more detail. The fact that Tie2 KO leads in part to the acquisition of an arterial phenotype is noteworthy and suggests something of an arteriovenous identity balance mechanism. Likewise, angiogenesis is an arterializing process. As such, the physiological downregulation of Tie2 in the angiogenic tip cell vasculature should/could likely be discussed in the context of the findings of this manuscript. Obviously, it is at the discretion of the authors to prioritize the discussion, but it is felt that the authors may in the present version of the manuscript not have discussed the most obvious and most exciting implications of their work.

Reviewer #3:

The study by Chu et al. characterizes the effect of Tie2 gene inactivation either using a ubiquitous or endothelial promoter to drive Cre expression during different stages of embryonic and postnatal stages. Interestingly, Tie2 deficiency leads to loss in vein formation or in stage-dependent maintenance of veins in different organs. The analyses are thorough and neatly presented. The study is ambitious, novel and of general interest.

1) The study shows that loss of Tie2 is accompanied by vascular abnormalities and a deficiency in vein formation. It is not entirely clear from the images that veins are missing entirely or if they are malformed. Please complement images with aSMA stainings (which should outline the arterial tree).

2) The author have previously described the iUCKO-/- mice in their paper in ATVB Jun;34(6):1221-30. Here, they have examined lymphatic vessel formation in the skin and show that deletion of Tie2 in neonate mice did not affect lymphatic vessel growth and maturation (Figure 8). I'm surprised that the lack of veins, as shown in the current study, has had no consequence for lymphatic vessel formation. The authors need to clarify and confirm that lymphatics are unaffected although veins are lacking.

3) In Figure 1, the authors examine embryos at E15.5 and in panel E, at E17.5. However, the embryos shown in the E panel seem considerably younger than the ones in the D panel? Magnification bars are missing for the embryos

4) In Figure 5, the authors show no difference in transcript levels of COUP-TFII when comparing retina and lung tissues from WT and iUCKO-/- mice. When analyzing protein levels (Figure 6) in lung tissue from mutant mice vs WT, the differences are now significant. Does loss of Tie2 lead to increased turnover of COUP-TFII? As COUP-TFII is a critical regulator of vein and lymphatic development, it is important to learn more about how it's regulated by Tie2. This has a bearing also on the (lack of?) effect of Tie2 deletion on lymphatic development.

[Editors’ note: what now follows is the decision letter after the authors submitted for further consideration.]

Thank you for submitting your work entitled "Angiopoietin receptor Tie2 is required for vein specification and maintenance via regulating COUP-TFII" for consideration by *eLife*. Your article has been reviewed by three peer reviewers, one of whom is a member of our Board of Reviewing Editors, and the evaluation has been overseen by a Senior Editor.

While the reviewers feel that your work is potentially very important, they were disappointed that you chose to argue points rather than making the necessary revisions elaborated in the original set of reviews. You will see from their comments, that none of the reviewers felt you adequately addressed their original concerns. Normally, *eLife* only allows for a single round of reviews. Given the importance of your work and the fact that all three reviewers have urged me to give you another opportunity to revise the paper, we are willing to give you a second chance to make the original required major revisions. I would like to stress that the reviewers are your advocates and trying to help you improve the paper. The reviewers’ comments from the second round of review are listed below, but I urge you to make the revisions requested in both the first and second round of review.

This will be your final opportunity to provide acceptable additional experimental evidence as required by the reviewers.

Reviewer #1:

Overall my feeling is that the authors could have a very nice story but they do not want to make an effort to make it more complete. Data on the mechanism of action remain the same, as poor as before.

I do not think that they can conclude that since COUP-TFII is reduced when Akt activation is inhibited (in HUVEC) Akt activation is the mechanism of action of Tie-2 in maintaining a venous identity. Although there is a partial inhibition of Akt (24% reduction) in Tie-2 deficient mice other pathways are likely implicated.

pDok- 2, for instance, seems more dramatically affected. It is sad that they do not want to make a relatively small effort to improve their paper.

The same applies to the relevant points raised such as the use of a model of ubiquitous inactivation of Tie-2 versus endothelial specific inactivation, or the lack of studies on the possible alterations of the lymphatics in absence of a correct venous development.

Reviewer #2:

The manuscript by Man Chu and coworkers is resubmitted after revision. That global deletion of Tie2 casus venous defects is convincingly demonstrated. I have the following comments:

1) A major concern with this study is the weak phenotype of the endothelial deletion compared to the ubiquitous deletion. The strong effects in the global deletion could be primary as well as secondary.

2) The question on how lack of veins affect lymphatics remains unanswered. The authors published in ATVB in 2014, that lymphatic vessel development is unaffected in neonate pups with the same global Tie2 deletion as described here. It is unexpected that the lack of veins would not affect lymphatic development.

3) Mechanistic in vivo data remain scarce.

Reviewer #3:

This reviewer is rather disappointed on how the authors have addressed the specific critique to the original submission of the manuscript. The idea of the peer review process is to initiate an iterative process aimed at improving the eventually published manuscript. As such, this reviewer had constructively raised a number of issues that he felt would help to improve the manuscript. Generally speaking, the authors have not accepted this invitation, but essentially argued away all of the reviewer's critique and suggestions. They thereby missed the opportunity to improve their manuscript (at least in response to the comments and suggestions made by this reviewer). For example, they did not even opt to replace the black IF images for the supposed Tie2 ECKO analyses. In the rebuttal, they argue at length about different recombination efficacies using the two different driver lines. Clearly, incomplete recombination and the resulting competition of WT and KO cells in long term experiments is key to the interpretation of conditional KO data. If the authors' recombination analyses are correct, then this reviewer would insist that the corresponding IF images should show mosaic Tie2 expression and not black images.

Despite this disappointment about the revised manuscript, this reviewer considers the manuscript of high quality and in principle deserving of publication. As such, if the authors are either not able or not willing to adequately address this reviewer's concern and suggestions, the manuscript should be published as is.

---

## [Author Response]

[Editors’ note: the author responses to the first round of peer review follow.]

*[…] Reviewer #1:*

*The paper by Man Chu et al. describes the effect of inactivation of Tie-2 expression on artero-venous differentiation in the mouse embryo, pups and cultured endothelial cells.*

As previously reported, Tie 2 null embryos die early during gestation for vascular and heart developmental problems. Here the authors show that Tie-2 deficient embryos and pups present a defective venous organization when Tie-2 expression is missing. In some organs and tissues, veins are absent along the arteries. In the retina there is a reduced horizontal growth of the vasculature and the presence of malformations along the length of the veins. In the skin, veins look tortuous. Arterial and venous markers are partially modified. In HUVEC silenced for Tie2 or lung and liver tissues Coup-TFII is reduced.

*Mechanistically, data suggest that Tie-2 may act through phosphorylation of Akt that in turn would increase COUP-TFII and venous differentiation.*

*Although the paper contains some interesting observations it is essentially descriptive. The major weakness is the lack of a detailed mechanistic analysis of how Tie-2 acts in inducing venous differentiation. Data on the role of Akt are suggestive but a marked reduction of COUP-TFII is detectable only in HUVEC KD for Tie-2 and is much less apparent in the tissues. Other published studies showed an increase in cultured cells of Akt and STAT 1 upon infection with Tie-2 gain of function mutant. In addition, inherited gain of function mutations of Tie-2 induce a strong venous endothelial cell proliferation and formation of large hemangiomas inhibited by Rapamycin (for instance Boscolo et al. J Clin invest). To my view the authors should extend their studies on the mechanism of action of Tie-2 also adding novel observations as compared to previously published literature.*

In this study we have found that Tie2 is required for vein specification and maintenance via Akt mediated regulation of COUP-TFII. Specifically, as shown in Figure 6, there is a significant reduction of COUP-TFII protein in tissues after the induced deletion of *Tie2*, and this is confirmed by the siRNA mediated knockdown of *Tie2* in HUVECs (Figure 6). On the other hand, the COUP-TFII level was increased by COMP-Ang1 stimulation to activate the Tie2/Akt pathway (Figure 6). Consistently, a significant reduction of COUP-TFII is achieved by the inhibition of PI3K/Akt using LY294002 or MK2206 (Figure 6). The decrease of COUP-TFII by the Akt inhibitor MK2206 could be reverted by the proteasome inhibitor MG132 (Figure 6). These findings indicate that Tie2 is important in venogenesis via Akt mediated stabilization of COUP-TFII.

The decrease of Akt activation after Tie2 deletion as shown in Figure 2 is consistent with the observation that there is an increase of Akt activation in cultured endothelial cells transduced with retroviral vectors overexpressing an activating Tie2 mutant (Tie2-L914F, Boscolo et al., J Clin invest. 2015; 125: 3491-3504). However, activation of Tie2 by its ligand Ang1 or VE-PTP inhibition, at least for a short period, has been shown to stabilize blood vasculature (Shen et al., J Clin Invest. 2014; 124: 4564-4576), while activating Tie2 mutations lead to venous malformations (Vikkula et al., 1996; 87: 1181-1190; Limaye et al., 2009; 41:118-124; Boscolo et al., J Clin invest. 2015; 125: 3491-3504). Mechanisms underlying the discrepancy require further investigation.

*Reviewer #2:*

*The authors of the manuscript, "Angiopoietin receptor Tie2 is required for vein specification and maintenance via regulating COUP-TFII" have used global and endothelial cell-specific conditional Tie2 knockout mouse models to demonstrate the critical role of Tie2 during vein development. The study is novel and addresses a fundamental question regarding the pathways (Tie2, Akt/PI3K and COUP-TFII) involved in forming venous endothelial identity and function. Although there is already a lot of circumstantial evidence in the published literature on vein-specific functions of Ang/Tie signaling, the manuscript makes an important point in showing this conclusively using definite genetic models. The study will be of interest for the broad readership of the journal. In further advancing their work, the authors should consider the following:*

*1) The major limitation in the study is the use of a ubiquitously expressed Cre deletor line (UBC-Cre^ERT2^) to knockout Tie2. Even though the authors have also used an endothelial specific Cre line (Cad5-Cre^ERT2^) to demonstrate the phenotypes in embryos and early postnatal retina, all the mechanistic studies are only performed using the UBC-Cre^ERT2^ line. This is a major concern especially because Tie2^-/iUCKO^ seems to show better Tie2 deletion as well as stronger phenotypes compared to Tie2^-/iECKO^ line. Is this solely due to different recombination efficacies of the different drivers or would this difference raise the question whether endothelial cell autonomous effects may not be the only cause of defective vein development in Tie2-deficient mice? Why did the authors opt to study different time points in Figure 1?*

The abnormal vein development was observed in skin of *Tie2^-/iECKO^* mice, and also in *Tie2^-/iUCKO^* but not in *Tie2* mutants with its deletion in blood cells (*Tie2^Flox/-^;Vav-iCre*), suggesting that it is an endothelial cell autonomous effect. Postnatally, the milder vascular effect observed in retina of *Tie2^-/iECKO^* mice may be mainly due to the lower deletion efficiency of *Tie2*. The remaining Tie2 mRNA level is approximately 10-20% in *Tie2^-/iUCKO^* mice (Figure 5- table supplement 1) and it is about 40% in *Tie2^-/iECKO^* mutants compared with the respective control mice (*Tie2^-/iECKO^*: 0.40 ± 0.10, n=4; *Tie2^+/iECKO^*: 1.0 ± 0.21, n=4; also included in the text).

The analysis of skin vasculature with *Tie2^-/iUCKO^* mice at the same stage (E15.5, as *Tie2^-/iECKO^* mice shown in Figure 1) was shown in Figure 1—figure supplement 2. To study the stage-dependent roles of Tie2 in vascular development, the Tie2 deletion was induced two days later (from E12.5-14.5) in *Tie2^-/iUCKO^* mice as shown in Figure 1. Interestingly, although the defective vein development exists in *Tie2^-/iUCKO^* mice, there is no bleeding and edema observed at E17.5.

*2) The immunofluorescence images show essentially black boxes in Figure 1 and in Figure 2 suggesting 100% deletion efficacy. Do the authors claim to have achieved 100% deletion of Tie2 in these experiments or why are these images not showing the expected mosaic staining pattern resulting from partial recombination? A quantitative analysis of the efficacy of Tie2 deletion in the different experiments should/needs to be included. Alternatively, qRT-PCR to show Tie2 expression should be done with retina and embryo tissues from Tie2^-/iUCKO^ (tissues and isolated endothelial cells). In Figure 2, the WB shows about 30-50% Tie2 in the Tie2^-/iECKO^ mice which is clearly not corresponding to the extremely weak or absent Tie2 staining shown in Figure 2.*

The deletion efficiency of Tie2 was also examined by western blot analysis and real-time RT-PCR, in addition to the immunostaining. The remaining levels of Tie2 transcripts in lung and retina of *Tie2^-/iUCKO^* mice were shown in Figure 5-table supplement 1. Analysis of Tie2 mRNA in lung of *Tie2^-/iECKO^* mice is now added as discussed above.

*3) The retina vasculature analysis at P11, P15 and P21 in Tie2^-/iUCKO^ mice showing vascular tuft formation is interesting. The same analysis should be included using the Tie2^-/iECKO^ mice.*

We did not observe the vascular tufts in retina of *Tie2^-/iECKO^* mice at P21, and this may be due to the lower efficiency of Tie2 gene deletion as discussed above. According to the recent work in the lab of Dr. Gou Young Koh (also the co-author of this paper), similar vascular defects were observed in retinas of *Tie2^-/iECKO^* mice when a higher efficiency of Tie2 deletion was achieved using a different VE-*Cadherin-Cre^ERT2^* mouse line.

*4) Although hemangioma-like vascular tufts were observed in Tie2^-/iUCKO^ mice at P21, the adult retina did not show any such malformations, whereas ear skin showed tortuous veins in adult mice. These data have been discussed rather insufficiently. The authors should comparatively analyze COUP-TFII transcript levels and other downstream players such as pAkt and pDok-2 in these two vascular beds in order to dissect the mechanisms, which help the retinal veins to "recover" but not the cutaneous veins. Likewise, partial recombination will lead to a mosaic vasculature of WT and KO cells. If such mosaic mice are traced over time, there is a good chance that the KO cells will be competed out by WT cells. As such, it is critical, particularly in the longer term experiments to carefully trace the percentage of cells with Tie2 deletion at the later time points of analysis.*

As shown in Figure 4, we have now added new data from the analysis of retinal vasculature in *Tie2^-/iUCKO^* and control mice at the adult stage (2.5 month-old). Similar defects with venous tortuosity was also observed in the retina of *Tie2^-/iUCKO^* mice. Consistently, there was little vascular growth to the second layer and no blood vessel growth into the third layer of retina in *Tie2^-/iUCKO^* mice. Furthermore, Tie2 transcript level remained low in *Tie2^-/iUCKO^* mice as analysed by the immunostaining (Figure 4) and real-time RT-PCR (*Tie2^-/iUCKO^*: 0.26 ± 0.14, n=5; Control: 1.0 ± 0.28, n=5). The data is now included in the text.

*5) The authors claim that Tie2 is required for the maintenance of venous EC identity. This is suggested by the observed phenotypes (albeit only moderately convincing). A systematic gene expression analysis of adult retina similar to Figure 5 (performed using P7 retina and lung) is missing. It is important to know whether the expression of venous markers is restored over time and also whether Tie2 expression is recovered in the adult mice as a result of incomplete Cre recombination efficiency and compensatory proliferation of Tie2-expressing EC (see above).*

As discussed above, Tie2 transcript level remained low in *Tie2^-/iUCKO^* mice at the adult stage, and retina was abnormally vascularized as shown in Figure 4. By real-time RT-PCR, we found that both venous and arterial genes are decreased in *Tie2^-/iUCKO^* mice compared with that of controls (EphB4, *Tie2^-/iUCKO^*: 0.52 ± 0.29, n=5; Control: 1.0 ± 0.18, n=5; APJ, *Tie2^-/iUCKO^*: 0.22 ± 0.22, n=5; Control: 1.0 ± 0.25, n=5; Ephrin B2, *Tie2^-/iUCKO^*: 0.71 ± 0.32, n=5; Control: 1.0 ± 0.23, n=5; Dll4, *Tie2^-/iUCKO^*: 0.70 ± 0.36, n=5; Control: 1.0 ± 0.21, n=5). This may be due to the fact that endothelial cell numbers are different between Tie2 knockout and littermate control mice due to the disruption of EC survival pathway.

*6) The authors are encouraged to reconsider their discussion. They conclude at the end of the discussion that the findings put a cautionary note on long term Tie2 targeting therapies, e.g., during tumor angiogenesis. This may not be the most important point of the study. In fact, nobody tries to target Tie2. Ang2 neutralizing strategies primarily work cooperatively with VEGF pathway targeting drugs based on their vascular normalization Tie2 gain-of-function effect. The same holds true for VE-PTP inhibitors or the recently published ABTAA antibody. What this reviewer finds much more exciting is the role of Tie2 on venogenesis. Venogenesis has mostly been considered as the default pathway with transcriptional programs driving lymphatic and arterial differentiation. The identification of Tie2 signaling as upstream regulator of COUP-TFII is a major finding that should likely be discussed in more detail. The fact that Tie2 KO leads in part to the acquisition of an arterial phenotype is noteworthy and suggests something of an arteriovenous identity balance mechanism. Likewise, angiogenesis is an arterializing process. As such, the physiological downregulation of Tie2 in the angiogenic tip cell vasculature should/could likely be discussed in the context of the findings of this manuscript. Obviously, it is at the discretion of the authors to prioritize the discussion, but it is felt that the authors may in the present version of the manuscript not have discussed the most obvious and most exciting implications of their work.*

The cautionary note in the Discussion is deleted. A new sentence has been added to the first paragraph of Discussion- “This implies that Tie2/Akt signaling may counterbalance VEGFR2/MAPK pathway in arteriovenous specification during the vascular development.”

*Reviewer #3:*

*The study by Chu et al. characterizes the effect of Tie2 gene inactivation either using a ubiquitous or endothelial promoter to drive Cre expression during different stages of embryonic and postnatal stages. Interestingly, Tie2 deficiency leads to loss in vein formation or in stage-dependent maintenance of veins in different organs. The analyses are thorough and neatly presented. The study is ambitious, novel and of general interest.*

*1) The study shows that loss of Tie2 is accompanied by vascular abnormalities and a deficiency in vein formation. It is not entirely clear from the images that veins are missing entirely or if they are malformed. Please complement images with aSMA stainings (which should outline the arterial tree).*

In embryos, skin veins were completely missing when the induced deletion of Tie2 gene started from E10.5. The absence of veins in the skin were confirmed by co-staining of PECAM-1 and Dll4 as shown in Figure 1—figure supplement 1.

*2) The author have previously described the iUCKO-/- mice in their paper in ATVB Jun;34(6):1221-30. Here, they have examined lymphatic vessel formation in the skin and show that deletion of Tie2 in neonate mice did not affect lymphatic vessel growth and maturation (Figure 8). I'm surprised that the lack of veins, as shown in the current study, has had no consequence for lymphatic vessel formation. The authors need to clarify and confirm that lymphatics are unaffected although veins are lacking.*

We have performed the analysis of lymphatic development with *Tie2^-/iUCKO^* and control mice during embryogenesis, and the data was not included in the paper. We found that there was abnormality about the lymphatic network in the skin analyzed at E17.5 (e.g. lymphatic vessel dilation). However, this could be secondary to the tissue edema resulting from the defective vein development. The detailed mechanism underlying the lymphatic defects in *Tie2^-/iUCKO^* mice during embryogenesis is still to be investigated.

*3) In Figure 1, the authors examine embryos at E15.5 and in panel E, at E17.5. However, the embryos shown in the E panel seem considerably younger than the ones in the D panel? Magnification bars are missing for the embryos*

The images in Figure 1 were taken using different magnification. Scale bars have now been included in the figure.

*4) In Figure 5, the authors show no difference in transcript levels of COUP-TFII when comparing retina and lung tissues from WT and iUCKO-/- mice. When analyzing protein levels (Figure 6) in lung tissue from mutant mice vs WT, the differences are now significant. Does loss of Tie2 lead to increased turnover of COUP-TFII? As COUP-TFII is a critical regulator of vein and lymphatic development, it is important to learn more about how it's regulated by Tie2. This has a bearing also on the (lack of?) effect of Tie2 deletion on lymphatic development.*

As shown in Figure 6, COMP-Ang1 stimulation could increase COUP-TFII level in cultured ECs, which is consistent with the observation that induced Tie2 deletion or siRNA mediated knockdown of Tie2 led to the decrease of COUP-TFII protein level (Figure 6). Furthermore, the decrease of COUP-TFII by PI3K/Akt inhibition could be reverted by the proteasome inhibitor MG132. The findings indicate that Tie2 mediated Akt activation is important for the stabilization of COUP-TFII protein.

As discussed above, lymphatic abnormalities were observed in *Tie2^-/iUCKO^* mice. However, as Tie2 is lowly expressed by lymphatic endothelium, it remains to be investigated whether it is a primary or secondary defects resulting from Tie2 insufficiency.

[Editors’ note: the author responses to the re-review follow.]

*[..] While the reviewers feel that your work is potentially very important, they were disappointed that you chose to argue points rather than making the necessary revisions elaborated in the original set of reviews. You will see from their comments, that none of the reviewers felt you adequately addressed their original concerns. Normally, eLife only allows for a single round of reviews. Given the importance of your work and the fact that all three reviewers have urged me to give you another opportunity to revise the paper, we are willing to give you a second chance to make the original required major revisions. I would like to stress that the reviewers are your advocates and trying to help you improve the paper. The reviewers’ comments from the second round of review are listed below, but I urge you to make the revisions requested in both the first and second round of review.*

*This will be your final opportunity to provide acceptable additional experimental evidence as required by the reviewers.*

*Reviewer #1:*

*Overall my feeling is that the authors could have a very nice story but they do not want to make an effort to make it more complete. Data on the mechanism of action remain the same, as poor as before.*

*I do not think that they can conclude that since COUPTFII is reduced when Akt activation is inhibited (in HUVEC) Akt activation is the mechanism of action of Tie-2 in maintaining a venous identity. Although there is a partial inhibition of Akt (24% reduction) in Tie-2 deficient mice other pathways are likely implicated.*

We agree with the reviewer that other downstream pathways, in addition to Akt mediated signaling, may also be involved in the venous abnormality when Tie2 mediated signaling was absent or insufficient. Considering that Akt is downstream of many signaling pathways and that it is also activated in many cell types, it is significant that we observed the decrease of pAkt in the total lung lysates after Tie2 deletion (Tek^-/iUCKO^ mice with gene targeting efficiency approximately 87% in lung as shown in Table 1; tamoxifen administration from P1-4 and tissues collected for analysis at P7). The finding was supported by the biochemical analysis with in vitro cultured endothelial cells. Firstly, consistent with the results from Tie2 knockdown by siRNA, the COUP-TFII level was increased by COMP-Ang1 stimulation to activate the Tie2/Akt pathway (Figure 6). Secondly, a significant reduction of COUP-TFII was achieved by the inhibition of Akt using MK2206 (Figure 6), which validated the results from PI3K inhibition by LY294002 (Figure 6). The decrease of COUP-TFII by the Akt inhibition could be reverted by the proteasome inhibitor MG132 (Figure 6). The findings suggest that Tie2 regulates COUP-TFII protein stability, which may act via Akt mediated signaling. We have revised the text in the Results section by including these data. We have also revised the Discussion section to reflect the reviewer’s excellent view, pointing out that in addition to Akt, other factors / pathways may also participate in mediating Tie2 signals in venous specification.

*pDok- 2, for instance, seems more dramatically affected. It is sad that they do not want to make a relatively small effort to improve their paper.*

We thank the reviewer to point this out. Indeed both total Dok2 and phospho-Dok2 level were significantly reduced in mice with the induced deletion of Tie2 (Figure 2). So far, there is little information available about the role of Dok2 in blood vascular development. In experiments using HEK 293T or endothelial cells overexpressing Dok2, it was found that Dok2, via directly interacting with Tie2, mediated Tie2 pathway in endothelial cell migration (ref. 29, 30). We examined the endogenous level of Dok-2 in HUVECs by western blot analysis and RT-PCR and found that Dok-2 was lowly expressed in cultured endothelial cells. We acknowledge the reviewer’s comment and have revised the Discussion by pointing out that further studies are important to analyze Dok-2 by employing genetically engineered mouse models targeting *Dok2*.

*The same considering the relevant points you raised such as the use of a model of ubiquitous inactivation of Tie-2 versus endothelial specific inactivation, or the lack of studies on the possible alterations of the lymphatics in absence of a correct venous development.*

For the embryo studies, both ubiquitous and EC specific deletion of *Tie2* were employed. Disruption of vein development was observed in Tek^-/iECKO^ (*Tek^Flox/-^/Cdh5-Cre^ERT2^)* as shown Figure 1, which was similar to that of Tek^-/iUCKO^
*(Tek^Flox/-^/Ubc-Cre^ERT2^*, Figure 1). In postnatal studies, although a significant decrease of retinal vascularization was also observed in Tek^-/iECKO^ mutants (Figure 2), we did not observe the vascular tuft formation in the retina of Tek^-/iECKO^ mice (data not shown). This may result from the lower Tie2 deletion efficiency in Tek^-/iECKO^ mice than that of Tek^-/iUCKO^ mice (as shown in Figure 2). However, we do not exclude the possibility that Tie2 expression by other non-endothelial cells may contribute to the vascular phenotype in Tek^-/iUCKO^ mutant mice, especially at postnatal stages. We have revised the Discussion section to point this out.

We have performed new experiments to analyze the lymphatic vessels in Tek^-/iECKO^ mutant mice. The results have now been included (Figure 1—figure supplement 3), and we have also revised the text in the Results section.

*Reviewer #2:*

*The manuscript by Man Chu and coworkers is resubmitted after revision. That global deletion of Tie2 casus venous defects is convincingly demonstrated. I have the following comments:*

*1) A major concern with this study is the weak phenotype of the endothelial deletion compared to the ubiquitous deletion. The strong effects in the global deletion could be primary as well as secondary.*

We also thank the reviewer to point this out. As discussed above, based on the embryo studies, the disruption of vein development is likely to be a cell-autonomous effect resulting from the deletion of Tie2 in endothelial cells (Figure 1). However, as discussed above, we cannot exclude the possibility that Tie2 deletion in non-endothelial cells may contribute to the vascular phenotype in Tek^-/iUCKO^ mutant mice, especially at postnatal stages. This requires further investigation. We have revised the Discussion section to point this out.

*2) The question on how lack of veins affect lymphatics remains unanswered. The authors published in ATVB in 2014, that lymphatic vessel development is unaffected in neonate pups with the same global Tie2 deletion as described here. It is unexpected that the lack of veins would not affect lymphatic development.*

This is an important question and we apologize for missing your point in the previous revision. In mammals, lymphatic vessels originate mainly from veins during embryonic development. It is therefore interesting to find out whether lymphatic development is altered in Tie2 knockout mice when the vein formation is disrupted. We have now performed new experiments to examine the lymphatic vessels in Tek^-/iECKO^ (*Tek^Flox/-^/Cdh5-Cre^ERT2^)* mice during embryogenesis. Lymphatic vessels were detected in Tek^-/iECKO^ mice but became dilated (Figure 1—figure supplement 3). This may be secondary to tissue edema resulting from the impairment of blood vasculature. However, it is also possible that Tie2 may have a role at earlier stages of lymphatic development. We have revised the text in the Results section.

*3) Mechanistic in vivo data remain scarce.*

We agree with the reviewer that further in vivo mechanistic studies are required for validating and fully elucidating Tie2 downstream mediators in the regulation of venous formation and maintenance. This may include the rescue experiments with Tie2 mutant mice as well as the generation of new genetic mouse models targeting the signal mediators downstream of Tie2 pathway (e.g. Dok-2). We have revised the Discussion section to reflect the reviewer’s valid point.

*Reviewer #3:*

*This reviewer is rather disappointed on how the authors have addressed the specific critique to the original submission of the manuscript. The idea of the peer review process is to initiate an iterative process aimed at improving the eventually published manuscript. As such, this reviewer had constructively raised a number of issues that he felt would help to improve the manuscript. Generally speaking, the authors have not accepted this invitation, but essentially argued away all of the reviewer's critique and suggestions. They thereby missed the opportunity to improve their manuscript (at least in response to the comments and suggestions made by this reviewer). For example, they did not even opt to replace the black IF images for the supposed Tie2 ECKO analyses. In the rebuttal, they argue at length about different recombination efficacies using the two different driver lines. Clearly, incomplete recombination and the resulting competition of WT and KO cells in long term experiments is key to the interpretation of conditional KO data. If the authors' recombination analyses are correct, then this reviewer would insist that the corresponding IF images should show mosaic Tie2 expression and not black images.*

We now have replaced the IF images for Tek^-/iECKO^ mice (Figure 1). In addition, to further visualize the remaining Tie2 in Tek^-/iECKO^ mice, we have performed new experiments with the mTmG allele bred into the Tie2 mutant mice. The partial Tie2 deletion was indicated by the mosaic GFP expression in the compound genetic mouse model (Tek^-/iECKO^;mTmG; Figure 2—figure supplement 1). We have also revised the text in the Results section.

Despite this disappointment about the revised manuscript, this reviewer considers the manuscript of high quality and in principle deserving of publication. As such, if the authors are either not able or not willing to adequately address this reviewer's concern and suggestions, the manuscript should be published as is.

We appreciate the highly positive comment and encouragement.